# Effect of Muscle Fibre Type on the Fatty Acids Profile and Lipid Oxidation of Dry-Cured Venison SM (*semimembranosus*) Muscle

**DOI:** 10.3390/foods11142052

**Published:** 2022-07-11

**Authors:** Joanna Żochowska-Kujawska, Marek Kotowicz, Małgorzata Sobczak, Sławomir Lisiecki

**Affiliations:** Department of Meat Technology, Faculty of Food Sciences and Fisheries, West Pomeranian University of Technology in Szczecin, Kazimierza Królewicza Str. 4, 71550 Szczecin, Poland; marek.kotowicz@zut.edu.pl (M.K.); malgorzata.sobczak@zut.edu.pl (M.S.); slawomir.lisiecki@zut.edu.pl (S.L.)

**Keywords:** venison, dry-cured muscle, histochemistry, fatty acids, lipids oxidation

## Abstract

The aim of the study was to describe the effect of fibre type on the fatty acid profile and lipid oxidation observed in dry-cured ham produced from individual *semimembranosus* venison (roe-deer, fallow deer, deer and wild boar) muscles. The results indicated that wild boar meat was characterised by the highest percentage of IA fibres and it contained the higher percentage of MUFA, but a low of PUFA and SFA, and was characterised by a(n-6)/(n-3) ratio lower than in the case of deer meat and greater susceptibility to oxidative changes. The highest percentage of SFA, and the lowest of MUFA and PUFA, was recorded in fallow deer meat, which was also characterised by the highest percentage of white fibres. The curing and drying processes increased the percentage share of SFA and the susceptibility of muscle lipids to oxidation, decreased the percentage of PUFA, and caused insignificant changes in the (n-6)/(n-3) ratio of fatty acids. The products were also characterised by a low amount of fat.

## 1. Introduction

The characteristics of dry-cured hams are shaped by the selection of appropriate processing conditions [1], raw materials of appropriate quality [2,3] or the amount and composition of adipose tissue [3,4]. As shown in the literature, the appropriate selection of the raw material, taking into account such factors as species, breed, sex, breeding, feeding and slaughter conditions, determines whether the finished product will be of high quality [4,5,6,7].

Pork is the basic raw material for the production of cured meats [8,9]. In many regions of the world, beef, goat, horse and other exotic species of game and farm animals are also used for the production of traditional dry-cured products, although information on the technology of processing of these types of meat is negligible [10,11]. A traditional product, available on the Italian market in many varieties, is bresaola, made of beef or horse meat [10], donkey meat [12] or bresaola di Cervo, which is made of game [11,13]. Spanish cecina is made of horse or beef meat and is similar to the biltong product characteristic of the regions of South Africa, made from the meat of exotic animals (antelope, gazelle, crocodile) or, more traditionally, from beef [14]. For the production of dry-cured hams, camel meat is also used [15], or even alpaca meat [16].

In recent years in Europe, there has been an increase in consumer interest in wild game meat as an alternative to pork and beef, mainly due to its relatively high nutritional value. New ranges of fermented and dried products from various species of game are beginning to appear on the market [11]. Despite the increase in popularity of this raw material, there is still little research to compare the quality of both meat and meat products produced from different species of game [17].

An important aspect in assessing the quality of raw material from wild game is taking into account the histochemical profile of the meat. According to the research of Żochowska-Kujawska et al. [18], a greater share of oxygen (red) fibres gives game greater hardness and poorer chewiness [19,20,21,22] and also less susceptibility to the massaging [23] or salting process [21] or requires additional tenderising, e.g., during marinating [22], in order to obtain the desired reduction of hardness and appropriate sensory or nutritional quality. Thus, the aim of the present study was to quantify the effects of fibre type on the fatty acid profile and lipid oxidation of dry-cured meat produced from individual roe-deer, fallow-deer, deer and wild boar’s m. semimembranosus (SM) muscles.

## 2. Materials and Methods

### 2.1. Harvesting of Hunting Animals

A total of 48 carcasses from roe deer (Capreolus capreolus), fallow deer (Dama dama), deer (Cervus elaphus) bucks and wild boar (Sus scrofa) males (twelve carcasses in each group) were used for the experiment. The carcasses selected from among the hunted animals weighed 14.7 kg (standard deviation = 0.8), 54.1 kg (SD = 1.7), 97.7 kg (SD = 4.2) and 73.2 kg (SD = 3.9), while their ages were approximately 24–30 months, respectively. The age of hunting animals was determined by tooth wear and replacement on the jaw teeth [24]. Animals were given unlimited access to a pasture located in the forests of the West Pomeranian Voivodeship (district of the city of Szczecin), which mostly includes grass, herbs, trees and brush. One-shot animals were harvested by experienced and licenced hunters during a winter hunting season (from October to February) within a closed area in the forest in the best possible conditions so as to minimise the stress [25]. Animals were bled by cutting the major vessels of the throat, and then immediately exsanguinated, weighed and tagged with an identification number.

### 2.2. Dressing and Removal of Muscles

For the purposes of histochemistry analysis, three samples of 1 × 1 × 0.5 cm were taken from the mid-part of the semimembranosus (SM) muscle of each carcass (45–60 min after being shot), then immediately frozen in liquid nitrogen and stored at −80 °C. The carcasses were transported to the laboratory facilities at the West Pomeranian University of Technology in Szczecin. Approximately 44-h post mortem, carcasses were taken from the cool room to be skinned, halved and jointed according to the standards of the Polish Department of Hunting and used to obtain 96 hams (24 hams from each species of animal). Each cut was deboned, and cleaned of external fat. The SM muscles of normal pH were dissected out of each of the hams. Before the salting stage, each muscle was shaped into a block by trimming off the edges.

### 2.3. Muscle Processing

The muscles obtained as described above were divided in a completely randomised design into three groups of eight SM and BF muscles of each species of animal. Each group of muscles was cured with 6% (*w*/*w*) of salt mixture, and the salting mixture contained 150 ppm of NaNO2 as a curing agent and NaCl as a salt base. The process conditions of production were determined in a previous experiment [21]. Thus, the salting phase was carried out at 3 ± 1 °C and 90% relative humidity (RH) for 2 days in a TCS-350 climatic chamber (Klimatest, Wrocław, Poland) until there was no salt visible on the muscle surface. After salting, all the muscles were held at 4–6 °C and 90% relative humidity for the next 2 days. At the post-salting stage, the muscles were hung in the chamber, while the temperature was raised from 6 to 15 °C for 10 days, and the relative humidity was progressively reduced to 75%. Subsequently, the muscles were ripened and dried in a chamber at 18–20 °C and 70–65% RH for the next 14 days. Then, the dry-cured muscles were vacuum-packed, and stored at 10 °C for about 24 h before the analysis. The entire process took 29 days.

### 2.4. Myofibre Classification and Measurements

Muscle fibre type measurements were obtained from muscle samples frozen in liquid nitrogen and cut at −24 °C with a HM 505 EV cryostat. The cuts (10 μm) were placed on glass slides, stained using the myosin ATP-ase method [26], and classified into three groups: type I (slow twitch, oxidative), type IIA (fast twitch, oxidative-glycolitic), and type IIB (fast twitch, glycolytic) [27,28]. The stained sections were examined with an image analysis system with the use of appropriate software (Multi Scan Base v.13). The percentage of type I, type IIA, and type IIB per each muscle fibre bundle was calculated with more than 10 bundles being examined for a muscle sample. A magnification of 100× was applied. The samples were analysed in duplicate.

### 2.5. Fat Content

Fat content was determined according to the AOAC (2005) procedure [29].

### 2.6. Fatty Acids Determination

#### 2.6.1. Lipid Extraction

The intramuscular fat from raw, post-salted and dry-cured SM muscles of all tested species of animals was extracted [30] using a ground sample of meat (5 g) which was separately homogenised using chloroform: methanol (2:1; *v*/*v*) solution. The extraction mixture contained 0.001% (*w*/*v*) of butylated hydroxytoluene as an antioxidant. The organic solvent was evaporated under a stream of nitrogen. The crude lipid extracts were then saponified with a 0.5 mol KOH solution in methanol. Afterward, the methyl esters of fatty acids (FAMEs) were prepared by transesterification with boron trifluoride (BF3) solution in methanol according to the AOCS official method Ce 2–66 [31].

#### 2.6.2. Fatty Acid Analysis

Fatty Acid Methyl Esters (FAME) were prepared according to the AOCS method [31]. GC analysis of FAME was carried out in the Agilent model 7890A instrument equipped with a split/splitless injector, MSD and a column, SPTM column: 2560, 100 × 0.25 mm ID, 0.20 µm film, catalogue number 24056. The initial temperature of the column was 145 °C, the injection port temperature was 220 °C, the detector temperature was 220 °C, the initial time was 5 min, the temperature increment was 4 °C/min, the final temperature was 220 °C and the total time of analysis was 45 min. Carrier gas helium: constant flow rate of 1.2 cm^3^/min, split ratio 1:50. Interpretation of chromatograms was made by comparing the retention times and the mass spectra of individual FAMEs of the examined sample with the retention times and mass spectra of the respective Sigma FAME standards (Lipid Standard). The results were recorded and processed using ChemStation (E.01.00) software and thus the fatty acids (mg/100 g of muscle tissue) as well as the n-6/n-3 fatty acid ratio were calculated. As an internal standard, C 19:0 was used. Only FAME representing > 0.01% of total FAME was included in the results.

### 2.7. Lipid Oxidation

The degree of lipid oxidation was determined according to the methodology presented by Buege & Aust [32]. Immediately prior to testing, TBA was prepared by mixing 30 mL of solution A (375 mg of thiobarbituric acid (TBA) dissolved in 30 mL of hot water) and 70 mL of solution B (15 g of trichloroacetic acid (TCA) dissolved in 67.9 mL of water) and 2.1 mL of 10 M HCl. A 5 g sample was homogenised with 25 mL of TBA, then heated for 10 min in boiling water until a pink colour was obtained. The mixture was then cooled under running water and centrifuged at 5500× *g* for 25 min. The supernatant absorbance was determined on a SEMCO spectrocolorimeter at a wavelength λ = 535 nm in a cuvette with a layer thickness of 1 cm. The amount of lipid peroxidation products reacting with thiobarbituric acid (TBA-RS) was calculated from the absorbance value using the molar absorption coefficient (ε = 1.56 × 105 M − 1 × cm − 1) and expressed in milligrams of malondialdehyde (MDA) per 1 kg of sample weight. TBA-RS concentration determinations were made for each variant in duplicate.

### 2.8. pH Measurement

Muscle pH was measured using a portable pH metre (CP-461, Elmetron, Zabrze, Poland) equipped with a pH penetration probe and an automatic temperature compensation probe. Before each measurement, the pH metre was calibrated with standard buffer solutions of pH 7.0 and pH 4.0 (POCH S.A., Gliwice, Poland) stored at room temperature.

### 2.9. Statistical Analyses

The instrumental measurement data were analysed statistically with the single effects given by species and the fixed effects by species and production phase, and their interaction [33]. The mean values and standard error of means (SEM) for each muscle (raw and dry cured) as well as the differences in the histochemistry and some chemical properties between venison muscles and production phase using RIR-Tukey test are presented in Table 1, Table 2 and Table 3.

## 3. Results and Discussion

The study compares the histochemical profile (percentage of three types of muscle fibres) of the SM muscles of four game species (roe deer, fallow deer, deer and wild boar). The data in Table 1 shows that there was a variation between the examined muscles in terms of the share of three types of muscle fibres (I, IIA and IIB), and species was the factor that significantly differentiated the game muscles as a raw material for the production of dry-cured meats. The lowest number of red fibres was found in the meat of roe deer and fallow deer, the highest in the meat of deer and wild boar. Fallow deer meat had the highest percentage of white fibres, and deer meat had the highest percentage of IIA fibres.

There is little data in the literature on the simultaneous comparison of the histochemical muscle profiles of different species of wild animals, with the majority of them describing the histochemical state of the meat of wild boar, deer, or fallow deer [19,23,34,35]. Consequently, it is difficult to relate these results to the data presented in other studies. Interpretation of the results is also hindered by the fact that despite the genetically defined composition of fibres in the muscles [36], in response to many external (environmental) factors, their properties may change dynamically and adapt to new desired functions in the body [37]. Thus, the interspecific variation in the histochemical profile of venison may be the result of both natural differences between the studied species of animals and the current biochemical state of the muscle fibres. Under conditions of greater physical activity, type IIA (intermediate) fibres easily transform into type I or IIB fibres [38,39,40], which may increase the content of red fibres in the muscles of these species of animals, which are characterised by increased motor activity. Therefore, it can be assumed that the reason for the higher percentage of red fibres in the meat of wild boar than in the meat of roe deer, fallow deer and deer is the more active lifestyle of these animals [24].

There was a significant effect of the species on the fat content, fatty acid composition and the value of the TBA-RS coefficient (Table 2 and Table 3). Deer meat, with the highest share of PUFA (about 46% of total fatty acids), as well as roe deer meat (44.6%), were characterised by a more favourable composition of fatty acids. The highest amount of SFA (50.8% of total fatty acids), the lowest amount of MUFA and PUFA (12.1 and 37.6%, respectively), as well as a lower value of the (n-6)/(n-3) ratio were recorded for fallow deer meat. The highest fat content was found in wild boar and the lowest in fallow deer meat (Table 2).

It seems that the greater share of MUFA in venison meat and the higher TBA-RS ratio can be attributed to the greater percentage of red fibres it contains. In turn, a higher percentage of white fibres in the muscle was associated with a higher content of SFA. Moreover, Hernández et al. [41] and Andres et al. [42] ascribe a higher content of MUFA to pig muscles, which are dominated by oxidative fibres. On the other hand, there are studies whose authors did not confirm a significant effect of the type of muscle metabolism on the fatty acid profile of triglycerides [43,44], linking it rather with the influence of nutrition [45,46] or interbreeding [42].

In turn, a higher value of the TBA-RS index in muscles with a higher percentage of red fibres can be explained by a greater number of mitochondria in them, and consequently, a greater number of membranes being the main source of phospholipids [43], because—as shown by Andres et al. [42]—muscle sensitivity to oxidative changes depends both on the presence of compounds with antioxidant activity or pro-oxidative substances (mainly myoglobin), and on the total amount of phospholipids and PUFAs in the muscles. The reason for the differences in the values of the TBA-RS coefficient between the meat of the studied species of game could be the differences in the content of mineral salts in this raw material, especially copper, which is a catalyst for lipid oxidation, the high level of which is a characteristic feature of free-living animals, especially wild boars [34].

Apart from the above-mentioned influence of the histochemical profile, another cause of the interspecies differences in the quality of venison meat found during the study may be, among others, a diet associated mainly with a preference for a certain type of food, which, in turn, determines its nutritional value. Game has been shown to have a generally low fat content, which is also confirmed by data in the literature [47,48,49]. However, it seems that when assessing the nutritional value of the meat raw material, it is more important to estimate the “quality” of the fatty acid composition, e.g., by assessing the (n-6)/(n-3) ratio or the PUFA/SFA ratio (P/S), which, according to WHO recommendations, should be below 4.0 and above 0.4, respectively [50]. It turns out that the meat of all tested game species meets the requirements for functional food, as the value of the (n-6)/(n-3) ratio in the case of venison was 3.71–3.87 and was generally the highest in the case of red deer and roe deer and lower for fallow deer and wild boar. It can be seen that deer meat has the highest P/S value (P/S = 1.2), followed by wild boar and roe deer; the lowest value of the index is typical for fallow deer meat (P/S = 0.7). Since recently, attention has been paid to the presence of MUFA in the diet as those acids whose task is to reduce the risk of cardiovascular diseases [50], it should be emphasised that among the tested game species, the best source of these acids is wild boar and deer; fallow deer meat has the lowest MUFA and the highest SFA. Of course, venison should not be regarded as a valuable source of fat and PUFA, which it contains little compared to fish meat [50].

The observed differences in the fatty acid composition can be ascribed, as mentioned above, to the different diets of wild animals. Wild boar, despite being an omnivorous animal, prefer plant food, especially cultivated plants, but also acorns and chestnuts [24]. As a consequence, this meat has a different chemical composition, e.g., a higher content of MUFA and a lower content of SFA, compared to deer meat, in the case of which the basis of the diet is primarily grasses and plant shoots. In addition, Cava et al. [45], Cava et al. [46], Ruiz et al. [51] and Andres et al. [42] claim that the consumption of large amounts of acorns by Iberian pigs, which is also characteristic of wild boar, has a positive effect on changes in the fatty acid profile of their meat. In turn, a low (n-6)/(n-3) ratio in the meat of ruminants can be attributed to the high consumption of linolenic acid (C18:3) found in grass [50,52] as well as efficient hydrolysis of unsaturated acids from animal food by microorganisms of their gastrointestinal tract [53].

The curing and drying processes caused changes in the quality of muscles in all tested groups of animals, while both the phase of the production process and the type of meat were factors that differentiated the examined quality parameters (Table 2). It is known from the literature that the result of the proteolytic and lipolytic activity of endogenous enzymes are intense biochemical changes, leading to significant changes in the nutritional value of dry-cured products [54]. In assessing the nutritional value of dry-cured meats, the quality of fat related to the composition of fatty acids is an important indicator. As shown in the study, the curing and drying of venison meat, especially in the final stages, resulted in a decrease in the percentage of PUFA content (of about 5–20% depending on the type of meat and type of meat) and an increase in the amount of SFA (by about 14–29%) in relation to the raw material, despite the fact that, of course, the amount of fatty acids (mg/100 g) increased, which was a consequence of the increase in fat content caused by dehydration. However, a lower effect of drying on the percentage of MUFA acids was found, the amount of which was only slightly lower at the end of the production process than in the raw material. When assessing the quality of the finished product it is important that the production process, although its phase was considered a significant factor, had only a slight impact on the changes in the value of the (n-6)/(n-3) ratio, which in the case of the finished product may be considered favourable and close to the value of the raw material (Table 2).

The effect of drying time on changes in the fatty acid profile presented in the study and in the literature is ambiguous. It seems that one of the factors determining the nature of these changes is the type of raw material. For example, in curing and drying the horse tenderloin “cecina”, Lorenzo [55] found that the result of advanced lipolysis is both an increase in the amount of SFA and MUFA as well as PUFA. In turn, Buscailhon et al. [56] in studying the French drying hams, and Hernández et al. [57] in examining the Spanish dry-cured smoked pork loin showed no significant differences in the proportions of fatty acids. Another explanation for the differences in fatty acid changes occurring during lipolytic processes between the results of the present study and those presented in the literature is the time of production. As indicated by Motilva et al. [58,59] and Buscailhon et al. [56], intense lipolytic processes are typical of the first 5–6 months of the production process. However, in turn, Antequera et al. [60] suggest that in long-drying Iberian hams, intense lipolysis can occur both during the post-curing phase and during drying.

The result of the oxidation of unsaturated fatty acids, which accompanies the drying of cured meat, is associated with the increase in the value of the TBA-RS index observed in this study, compared to the raw material (Table 3), which was a consequence of the appearance of secondary oxidation products [7]. It can be assumed that the amount of secondary lipid oxidation products appearing in the product, regardless of the stage of the production process, may be related to the histochemical profile of the venison meat. It has been shown that both the raw material and the product in which initially a higher percentage of red fibres were found have a higher value of the TBA-RS index. A possible explanation for this is the higher content of oxidizable phospholipids in red fibres [42]. It is worth adding that the value of the TBA-RS index depended on the time (phase) of production, which is also confirmed by the studies of other authors on the meat of farm animals [46,61]. It should be noted, however, that the production process of venison dry-cured meats by processing individual muscles, devoid of a protective layer of adipose tissue, was shorter than the process typical of long-drying cured products made of whole basic elements. Therefore, the use of a modified drying process, without the need to use a lower temperature and relative humidity typical of the final ripening phase of long-drying hams, was probably the reason for the lack of a decrease in the value of the TBA-RS index. As demonstrated by Antequera et al. [60] and Cava et al. [46], in the case of Iberian pig hams, the susceptibility of lipids to oxidation is higher in the initial stages of production, i.e., during curing and drying, and then diminishes after about 700 days.

The species of game was another factor that, apart from the production process phase, influenced the quality of dry-cured meats. It seems that deer and wild boar meat products had better nutritional value, as they had a higher content of MUFA and PUFA, and a lower content of SFA. It is worth emphasising that the interspecies differences in the selection of raw material for the production of dry-cured meats had a significant effect on its nutritional value. The method of processing the raw material did not generally cause a significant deterioration in the fatty acid profile of dry-cured meats, regardless of the type of meat used. However, it can be noticed that dry-cured wild boar and deer meat had a slightly “better” profile of fatty acids.

One of the reasons for the differences in the fatty acid profile of dry-cured meats produced from different species of game animals indicated in the study may be the previously described differences in the quality of the raw material, which are a consequence of the diet preferred by game. Moreover, Ventanas et al. [62] believe that a necessary condition for obtaining Iberico ham is the use of meat from pure breeds of pigs as well as providing them with natural food in the form of fresh grass and acorns or chestnuts [63], which causes an increase in the level of both MUFA acids and a decrease in SFA levels as well as a more favourable ratio of fatty acids compared to animals fed with concentrated feed. Perhaps this explains the different fatty acid composition of wild boar hams, but also deer, for which acorns and chestnuts are a delicacy, compared to other species of game studied, whose diet is based on grasses and cereals [24].

## 4. Conclusions

The results indicated that the species of game is a factor that has a significant impact on the differentiation of the histochemical profile of muscles. However, the histochemical properties of game meat may determine its culinary quality and the possibility of using it in processing for the production of dry-cured meats. The production process is a factor that varies the fatty acid composition and the susceptibility of meat to oxidation. The processing of this raw material in the curing and drying process causes changes in the quality of the raw material but does not significantly deteriorate the fat quality of dry-cured meats.

## Figures and Tables

**Table 1 foods-11-02052-t001:** Mean fibre type percentage of SM (*semimembranosus*) venison muscles.

Fibre Composition	Species							
	Roe Deer		Fallow Deer		Deer		Wild Boar	
	Mean	s.e.m.	Mean	s.e.m.	Mean	s.e.m.	Mean	s.e.m.
Fibre typeI (%)	$21.71^{a}$	1.12	$29.16^{b}$	1.02	$37.52^{c}$	2.21	$40.46^{c}$	1.05
Fibre type IIA (%)	$22.53^{c}$	0.97	$9.41^{a}$	0.75	$30.41^{d}$	1.11	$18.27^{b}$	0.87
Fibre type IIB (%)	$55.76^{c}$	1.74	$61.43^{d}$	1.56	$32.07^{a}$	1.40	$41.27^{b}$	1.37
*p*-values	S							
Fibre typeI (%)	*							
Fibre type IIA (%)	*							
Fibre type IIB (%)	**							

*^a^*^–*d*^ values with different letters in the same row (species effect—S) statistically different at *p* < 0.05; ns—not significant, * *p* ≤ 0.05, ** *p* ≤ 0.01.

**Table 2 foods-11-02052-t002:** Fat content (% of total wet weight) and fatty acids composition (mg/100g muscle tissue) of SM (*semimembranosus*) venison muscles.

Fatty AcidsComposition	Species							
	Roe Deer		Fallow Deer		Deer		Wild Boar	
	Mean	s.e.m.	Mean	s.e.m.	Mean	s.e.m.	Mean	s.e.m.
** *Raw Muscles* **								
SFA	$228.99^{cA}$	12.87	$126.36^{aA}$	8.76	$195.90^{bA}$	6.12	$376.62^{dA}$	13.10
MUFA	$91.68^{bA}$	11.52	$31.94^{aA}$	1.76	$120.66^{cA}$	2.76	$297.88^{dA}$	10.06
PUFA	$289.32^{bA}$	13.54	$121.68^{aA}$	5.48	$283.44^{bA}$	10.50	$495.50^{cA}$	12.05
n-6/n-3	$3.74^{aA}$	0.06	$3.87^{aC}$	0.09	$3.84^{aA}$	0.12	$3.71^{aA}$	0.08
Fat (%)	$0.61^{bA}$	0.05	$0.28^{aA}$	0.07	$0.60^{bA}$	0.09	$1.17^{cA}$	0.06
** *post-salting phase* **								
SFA	$371.95^{cB}$	19.01	$238.63^{aB}$	12.93	$278.01^{bA}$	18.72	$451.17^{dB}$	16.62
MUFA	$125.30^{bB}$	4.13	$52.72^{aB}$	2.55	$156.58_{1}^{cB}$	10.98	$343.17^{dB}$	3.29
PUFA	$362.75^{bB}$	17.46	$198.64^{aB}$	9.02	$355.42^{bB}$	16.67	$545.11^{cB}$	25.06
n-6/n-3	$3.69^{aA}$	0.05	$3.68^{aB}$	0.04	$3.79^{aA}$	0.12	$3.68^{aA}$	0.18
Fat (%)	$0.86^{bB}$	0.11	$0.49^{aB}$	0.06	$0.79^{bB}$	0.08	$1.34^{cB}$	0.03
** *final product* **								
SFA	$1342.46^{cC}$	60.60	$561.43^{aC}$	26.04	$920.24^{bC}$	29.89	$1475.28^{dC}$	42.51
MUFA	$368.91^{bC}$	48.93	$105.21^{aC}$	3.78	$505.20^{cC}$	15.07	$914.68^{dC}$	26.87
PUFA	$1068.63^{bC}$	66.16	$383.35^{aC}$	18.38	$1004.52^{bC}$	26.24	$1620.04^{cC}$	56.94
n-6/n-3	$3.63^{aA}$	0.05	$3.51^{aA}$	0.11	$3.72^{aA}$	0.10	$3.69^{aA}$	0.16
Fat (%)	$2.78^{bC}$	0.24	$1.05^{aC}$	0.09	$2.43^{bC}$	0.21	$4.01^{cC}$	0.17
*p*-values	S			P		S × P		
SFA	**			**		ns		
MUFA	**			*		ns		
PUFA	*			**		ns		
n-6/n-3	**			*		ns		
Fat (%)	*			**		ns		

*^a^*^–*d*^ values with different letters in the same row (species effect—S) statistically different at *p* < 0.05; *^A^*^–*C*^ values with different letters in the same row (phase effect—P) statistically different at *p* < 0.05, ns—not significant, * *p* ≤ 0.05, ** *p* ≤ 0.01.

**Table 3 foods-11-02052-t003:** Mean values of 2-Thiobarbituric acid-reactive substances—TBARS (mg MDA/kg) of raw and dry-cured SM (*semimembranosus*) venison muscles.

Production Phase	Species							
	Roe Deer		Fallow Deer		Deer		Wild Boar	
	Mean	s.e.m.	Mean	s.e.m.	Mean	s.e.m.	Mean	s.e.m.
raw muscles	$0.163^{aA}$	0.02	$0.129^{aA}$	0.03	$0.151^{aA}$	0.02	$0.230^{bA}$	0.02
post-salting phase	$0.398^{bB}$	0.02	$0.227^{aB}$	0.02	$0.402^{bB}$	0.04	$0.419^{bB}$	0.03
final product	$0.515^{bC}$	0.03	$0.335^{aC}$	0.02	$0.493^{bC}$	0.04	$0.608^{cC}$	0.01
*p*-values	S			P		S × P		
	ns			**		ns		

*^a^*^–*d*^ values with different letters in the same row (species effect—S) statistically different at *p* < 0.05; *^A^*^–*C*^ values with different letters in the same row (phase effect—P) statistically different at *p* < 0.05, ns—not significant, * *p* ≤ 0.05, ** *p* ≤ 0.01.

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
