# Peer review of "Effect of Muscle Fibre Type on the Fatty Acids Profile and Lipid Oxidation of Dry-Cured Venison SM (semimembranosus) Muscle"

_foods, 2022, doi:10.3390/foods11142052_

Round 1

Reviewer 1 Report

The study compare dry cured hams from different species. It is a relatively simple set-up but still covering interesting analysis and I only have few comments. 

Line 220 ff. You are discussion the health benefit of the fatty acids, but at the same time according to table 2, the fat content is very low. So how much meat do you need to eat before this fatty acids has an real importance on health? This consideration could be included in the discussion.

 You also reflect on the influence of feed on the fatty acid composition, but with this low fat content, I would expect the mail part to be phospholipids, are the composition of these are relatively conservative. So may the difference simply  is due to different species?

Author Response

Thank you very much for your positive review.

I agree with the Reviewer's suggestion that venison is not a good source of fat (this comment was included in the discussion - line 234-235), however, the fat content in this raw material is very variable, mainly depending on the availability of food in the place of living (availability of the habitat to arable fields, type of crops, hunting season), which I have shown in other studies and may vary from 0.5 to even 14-15%.

  • Lachowicz i in. 2008. Effects of wild boars meat of different season of shot addition on texture of finely ground model pork and beef sausages. Electronic Journal of Polish Agricultural Universities. Series Food Science and Technology, 11,2.
  • Żochowska-Kujawska J., Lachowicz K., Sobczak M. 2010. Utility for production of massaged products of selected wild boar muscles originating from wetlands and arable area. Meat Science, 85, 461-466.
  • Żochowska-Kujawska et al. 2010. Dressing percentage and the percentage of prime cuts in the carcasses of wild boars depending on the season and region of shooting and sex. Medycyna Weterynaryjna, 66 (5), 335-338.

I also carried out a similar experiment on the meat of roe deer slaughtered between November and December, which is relatively fatty material, and the observed relationships were consistent with those presented in the reviewed manuscript. Unfortunately, in this case, I only have data on this one species.

I fully agree that the nutritional properties are the result of species belonging, but I think that, just as with the influence of environmental conditions, it is possible to modify the percentage composition of muscle fiber types, and in the same way we can change the composition and proportions of fatty acids.

Reviewer 2 Report

Manuscript ID: foods-1801308

Effect of muscle fibre type on the fatty acids profile and lipid oxidation of dry-cured venison SM (semimembranosus) muscle

I think it's a very interesting and very important topic in the meat context and meat quality nowadays as regards.  

The manuscript describe the effect of fibre type on the fatty acids profile and lipid oxidation observed in dry-cured ham produced from individual semimembranosus venison (roe-deer, fallow deer, deer and wild boar) muscles.

The topic is of interest for the academics and for the industry because of the results obtained and because of the potential use of different type of meat and its application in field. There are some studies like this in literature, but not specific.  The research is well performed, the sampling and analysis were well done.

Statistical analysis was well performed

The conclusions are of interest

The manuscript is well written and easy to understand by readers. I believe that this manuscript does not need big changes but I think you can publish the manuscript after minor revision and an improvement of discussions.

Specific suggestions

Line 27: …finished product will be of high quality.

please cite

·         Ambrosio, R.L.; Smaldone, G.; Di Paolo, M.; Vollano, L.; Ceruso, M.; Anastasio, A.; Marrone, R. Effects of Different Levels of Inclusion of Apulo‐Calabrese Pig Meat on Microbiological, Physicochemical and Rheological Parameters of Salami during Ripening. Animals 2021, 11, 3060. https://doi.org/10.3390/ani11113060

Line 41-43:… from different species of game…

 please cite

·         M.F.Peruzy, N.Murru, G.Smaldone, Y.T.R.Proroga, D.Cristiano, A.Fioretti, A.Anastasio. Hygiene evaluation and microbiological hazards of hunted wild boar carcasses. Food Control, Volume 135, May 2022, 108782. https://doi.org/10.1016/j.foodcont.2021.108782

Line 116: according to AOAC Methods.

Please cite

·         Smaldone, G., Marrone, R., Vollano, L., Peruzy, M.F., Barone, C.M.A, Ambrosio, R.L., Anastasio, A, 2019.   Microbiological, rheological and physical-chemical characteristics of bovine meat subjected to a prolonged ageing period. Italian Journal of Food Safety, Volume 8, Issue 3, 131-136

Author Response

Thank you very much for your positive review.

All references are included in the manuscript.